# Glial Cell Line-Derived Neurotrophic Factor-Loaded CMCht/PAMAM Dendrimer Nanoparticles for Peripheral Nerve Repair

**DOI:** 10.3390/pharmaceutics14112408

**Published:** 2022-11-08

**Authors:** Ane Escobar, Mariana R. Carvalho, F. Raquel Maia, Rui L. Reis, Tiago H. Silva, Joaquim M. Oliveira

**Affiliations:** 13B’s Research Group, I3Bs—Research Institute on Biomaterials, Biodegradables and Biomimetics, Headquarters of the European Institute of Excellence on Tissue Engineering and Regenerative Medicine, University of Minho, AvePark, Parque de Ciência e Tecnologia, Zona Industrial da Gandra, 4805-017 Barco, Portugal; 2ICVS/3B’s—PT Government Associate Laboratory, 4710-057 Braga, Portugal

**Keywords:** peripheral nerve injury, peripheral nerve repair, nanoparticles, neurotrophic factor, dendrimers

## Abstract

(1) Background: Peripheral nerve injuries represent a major clinical challenge. If nerve ends retract, there is no spontaneous regeneration and grafts are required to proximate the nerve ends and give continuity to the nerve. (2) Methods: GDNF-loaded NPs were characterized physicochemically. For that, NPs stability at different pH’s was assessed, and GDNF release was studied through ELISA. In vitro studies are performed with Schwann cells, and the NPs are labeled with fluorescein-5(6)-isothiocyanate for uptake experiments with SH-SY5Y neural cells. (3) Results: GDNF-loaded NPs are stable in physiological conditions, releasing GDNF in a two-step profile, which is beneficial for nerve repair. Cell viability is improved after 1 day of culture, and the uptake is near 99.97% after 3 days of incubation. (4) Conclusions: The present work shows the efficiency of using CMCht/PAMAM NPs as a GDNF-release system to act on peripheral nerve regeneration.

## 1. Introduction

Peripheral neuropathies are a major source of disabilities and occur quite often. After an injury in the peripheral nervous system (PNS), the sensation can suffer distortion, and a loss in muscle mobility can occur [1]. The time-lapse between the injury and the regeneration determines the functional recovery of the muscle atrophy, which happens due to its loss of function while reinnervation occurs [2]. Nerve suturing is performed when the nerve ends do not retract. If they retract, a graft is required to join the proximal and distal segments of the injured nerve [1]. Autografts or artificial nerve constructs are used to guide and facilitate axonal growth. Nerve guidance conduits (NGCs) are tubular structures whose aim is to bridge the injured nerve ends, protect the regeneration from scar tissue formation and guide the regeneration from the proximal to the distal nerve stump [3].

Peripheral nerve repair happens due to the numerous growth factors (GFs) that Schwann cells (SCs) secrete and up-regulate [1], such as neurotrophin-3 or neurotrophin-4, the nerve growth factor, glial cell line-derived neurotrophic factor (GDNF), ciliary neurotrophic factor, and brain-derived neurotrophic factor [4,5]. Several works are directed to the exogenous administration of these GFs [6,7,8], as they are incorporated into artificial NGCs in order to promote peripheral nerve regeneration (PNR) to overcome long nerve gaps [9,10]. The main drawback of using ‘naked’ GFs administration is that they have a short biological half-life, the reason why they degrade fast, without improving nerve regeneration. However, if they are complex, or in this case, encapsulated in NPs, a slower and time-lasting release of the factor is possible to achieve [11,12,13,14,15,16].

To address this issue, in the present work, we are showing the synthesis of carboxymethyl chitosan/poly(amidoamine) (CMCht/PAMAM) dendrimer NPs loaded with GDNF for the exogenous administration of the GFs. The NPs’ physicochemical properties and stability are studied, and the GDNF release profile is evaluated through an Enzyme-Linked ImmunoSorbent Assay (ELISA). Immortalized SCs (iSCs) and SH-SY5Y neuroblastoma cells are used for biological evaluation, as they have a role [1] and act as an in vitro model for PNR [17], respectively. Cell viability is studied by incubating iSCs in the presence of the CMCht/PAMAM NPs without and with GDNF in order to observe the efficacy of the GDNF. SH-SY5Y cells are exposed to fluorescein-5(6)-isothiocyanate (FITC)-labeled NPs for cellular uptake studies, including cytometry and fluorescent imaging. Figure 1 shows a graphical illustration of the application of the studied CMCht/PAMAM NPs loaded with GDNF to release the GF to the lumen of NGCs.

## 2. Materials and Methods

CMCht/PAMAM NPs synthesis with GDNF: Carboxymethylchitosan (CMCht) with a degree of deacetylation of 80% and degree of substitution of 47% is synthesized by a chemical modification route of chitin (Sigma, Darmstadt, Germany) as described by Chen and Park [18].

Starburst poly(amidoamine) (PAMAM) carboxylic-acid-terminated dendrimers (PAMAM-CT generation 1.5, 20% (*w*/*v*) methanolic solution) with an ethylenediamine core are purchased (Sigma-Aldrich). CMCht/PAMAM dendrimer NPs are prepared in a step-wise manner, as previously described by Carvalho et al. [19].

CMCht/PAMAM dendrimer NPs are mixed with previously diluted Human recombinant GDNF (Peprotech, Waltham, MA, USA), 10 μg in 900 μL of distilled water, and then precipitated after the addition of an appropriate volume of saturated Na_2_CO_3_ (Aldrich, Darmstadt, Germany) solution and cold acetone (Pronalab, Tlalnepantla, Portugal). Precipitates are collected by filtration and dispersed in ultrapure water for dialysis for 48 h. CMCht/PAMAM dendrimer NPs are obtained by freezing the solution at 80 °C and freeze-drying (Telstar-Cryodos-80, Terrassa, Portugal) for up to 7 days to remove the solvent completely.

Field Emission Scanning Electron Microscopy (FE-SEM): The morphology of the NP is investigated by FE-SEM (Auriga Compact, Zeiss, Jena, Germany). For that, the NPs are stained with ammonium molybdate (Laborspirit, Santo Antão do Tojal, Portugal) and placed on carbon grids (TEM grids, Carbon Type-B 400M Cu, IESMAT, Madrid, Spain) for observation.

Dynamic light scattering: ζ-potential and particle size of the CMCht/PAMAM dendrimer NPs and GDNF-loaded CMCht/PAMAM dendrimer NPs are measured in a particle size analyzer (Zetasizer Nano ZS, Malvern Instruments, Malvern, UK). Particle size analyses are performed in an aqueous solution with a low concentration of NPs (0.1 mg mL^−1^) and using disposable sizing cuvettes. Electrophoretic determinations of ζ-potential are investigated using the universal dip cell in water.

NMR spectroscopy: Chemical structure is evaluated using H^1^NMR analysis. NPs (~7 mg) are dissolved in deuterated water (D_2_O), and the NMR spectra are obtained in a Mercury-400BB operating at a frequency of 399.9 MHz at 50 °C. The 1D 1H spectra are acquired using a 45° pulse, a spectral width of 6.3 kHz, and an acquisition time of 2.001 s.

Differential scanning calorimetry: DSC was performed in a DSC Q100 Model of T. A. INSTRUMENTS to study the thermal stability and changes in crystallinity over a range of temperatures. 3 mg of powder is placed in an aluminium pan, and a lid is crimped onto the pan. The pan is placed in the sample cell of the DSC module. Under N_2_ gas purge (50 mL min^−1^), the temperature is first equilibrated to −35 °C and increased at a rate of 10 °C min^−1^ until the material degrades.

pH stability of CMCht/PAMAM NPs with GDNF: NPs stability at different pH values is assessed by DLS size measurement. NPs are dissolved and sonicated in 10 mM PBS solution adjusting the pH with 1 M NaOH and HCl to values of 3, 5, 7, 8, 9, 10, and 12.

GDNF release for the CMCht/PAMAM NPs: The amount of GDNF release from the NPs was quantified by ELISA according to the manufacturer’s protocol. Human GDNF Duoset ELISA kit is purchased from R&D Systems (USA). NPs at a concentration of 5 mg mL^−1^ in 1 wt % BSA in PBS are kept in a bath at 37 °C under orbital agitation at 60 rpm, and three samples are prepared to measure at each time-point. After 15 min, 1 and 4 h and 2, 7, 14, and 30 days, the absorbance of the supernatant of the collected solutions is measured at 450 nm in a plate reader (Tecan, Spark 10M, Männedorf, Switzerland). With the standard supplied by the manufacturer, a calibration curve is performed to determine the GDNF concentration. 

Cell Culture of Immortalized human Schwann Cells and SH-SY5Y neuroblastoma Cells: Human iSCs (sNF96.2, ATCC) and SH-SY5Y neuroblastoma cell line (Sigma-Aldrich) are cultured in High Glucose DMEM with 1% sodium pyruvate and DMEM, respectively, both supplemented with 10% FBS and 1% penicillin/streptomycin. Cells’ medium is changed every 2–3 days, and they are kept at 37 °C and 5% CO_2_.

Cellular viability evaluation: The viability of iSCs is followed with Alamar Blue (AB), a dye that yields a fluorescent signal when incubated with metabolically active cells. Cells were cultured at a density of 104 cells well-1 and left to adhere for 24 h. Then, the medium is changed to medium supplemented with 0.1, 0.5, and 1 mg mL^−1^ of CMCht/PAMAM NPs without and with GDNF. No supplemented medium was used as a control. On days 1 and 3, iSCs were incubated for 3 h in 20% AB in a medium. Fluorescence is monitored at a 590 nm emission wavelength and excitation wavelength of 530 nm using an FL 600 Bio-Tek Instruments microplate reader. Three replicates are made of each sample. After the incubation time, PBS was used to rinse the AB reagent, and a fresh culture medium was added to continue the culture.

FITC-conjugated CMCht/PAMAM NPs uptake by SH-SY5Y cells: The cellular uptake at 24 and 72 h is studied by fluorescence microscope imaging and cytometry analysis. CMCht/PAMAM NPs without and with GDNF are first labeled with FITC. Briefly, 10 mg mL^−1^ of NPs are prepared in a carbonate-bicarbonate coupled buffer at pH 9.2, and a solution of FITC/DMSO is then added under agitation and left for reaction in the dark for 8 h. The amine groups of the CMCht are bonded covalently to the isocyanate group of the FITC (Sigma, Germany). The conjugates are dialyzed in ultrapure water for 24 h before freeze-drying.

Then, cells are seeded at a concentration of 2 × 10^4^ cells per well in a 24-well plate for fluorescence imaging and for cytometry at 2 × 10^5^ cells per well in a 6-well plate. After 24 h, the medium is renewed with medium supplemented with CMCht/PAMAM NPs without and with GDNF at 0.5 mg mL^−1^ and control cells are cultured in a complete medium. Three replicates are made of each sample.

Cellular uptake is visualized by fluorescence microscopy. For that, SH-SY5Y cells are washed with PBS and fixed with 4% formalin for 20 min at RT. After, fixed cells are washed three times with PBS, and 0.2% Triton X-100 in PBS is added to each well for cells’ permeabilization. Then, F-actin filaments are labeled with Texas Red-X phalloidin (Molecular Probes, Invitrogen, Waltham, MA, USA) and the nuclei with 4,6-diamidino-2-phenylindole, dilactate (DAPI blue, Molecular Probes, Oregon, UK). The cells are imaged by fluorescence microscopy (AxioImager Z1, Zeiss Inc., Oberkochen, Germany) to assess the internalization of the FITC-conjugated NPs.

Internalization quantification of the FITC-conjugated NPs is performed by flow cytometry analysis. Cells were detached using TrypleX (Life Technologies, Waltham, MA, USA) and transferred to cytometry tubes to be further analyzed in the FACSCalibur flow cytometer (BD Biosciences, Franklin Lakes, NJ, USA).

Statistical analysis: The statistical ANOVA analysis is performed in OriginPro 2016 software version b9.3.226. Fisher’s tests are performed to determine statistically significant differences with a *p* < 0.05.

## 3. Results

### 3.1. Physicochemical Characterization of GDNF-Loaded CMCht/PAMAM NPs

Particle size distribution in solution is assessed by DLS in aqueous solution, and intensity values are plotted in Figure 2A, revealing a size of 88.45 ± 20.13 nm and 115.00 ± 29.30 nm for CMCht/PAMAM NPs without and with GDNF, respectively. In dry conditions (Figure 2B), the architecture of the NPs can be depicted, and the polymer coating around the PAMAM core can be identified. The FE-SEM images also revealed that GDNF-loaded NPs were more compact with a more defined shape. ζ-potential results confirmed the negative surface charge of the NPs when dissolved in water, being of −40.00 ± 6.45 mV for CMCht/PAMAM NPs and −37.2 ± 3.73 mV CMCht/PAMAM NPs with GDNF, this decrease in the change might be due to the positive change of the GDNF, that reduces the overall negative ζ-potential of the NP.

The H^1^NMR spectra of the synthesized CMCht/PAMAM dendrimer NPs without and with GDNF are plotted in Figure 2C. A doublet at 1.24 ppm, singlet at 2.49 ppm, and multiplets from 3.1 to 3.73 ppm and 4.07 ppm are associated with the H^2^ protons, the ring methine protons (H^3^, H^4^, H^5^, and H^6^), and protons of –CH_2_COO– groups of the CMCht [20]. Peaks at 2.49 and 3.46 ppm appear to overlap as a result of the resonances of the CH_2_COO and CH_2_ protons of both CMCht and PAMAM [20,21]. Two peaks appear at around 2 to 2.2 ppm, which are associated with carbonyl α hydrogens [22], the second one is lower in GDNF-loaded CMCht/PAMAM dendrimer NPs due to the electrostatic interaction between the CMCht/PAMAM NPs and the GDNF. The highly intense chemical shift at around 4.65 ppm refers to the D_2_O [23], the solvent used for the measurements.

The thermal stability is assessed through DSC thermal stability analysis (Figure 2D). The glass transition temperature (Tg) is slightly lower when NPs contain GDNF, being −7.70 °C and −9.40 °C for CMCht/PAMAM NPs without and with GDNF, respectively. The thermodynamic crystallization temperature (TCT) is also lower for NPs with GDNF; 91.18 vs. 70.78 °C and the thermal decomposition point is the same for both NPs, being 257.86 vs. 257.64 °C. 

### 3.2. NPs Stability

In order to evaluate NPs stability in different environments, their size is evaluated in PBS 10 mM at pH values of 3, 5, 7, 7.4, 8, 9, 10, and 12. DLS measurements are performed (Figure 3A), and the largest frequent radii size and the average radii are plotted (Figure 3B) to evaluate their stability. In most cases, the highest frequent peak values, plotted in blue, are not similar to the global average in red, which means large aggregates are formed with a large size that cannot be detected in the DLS. At alkaline pH values, the average radii size is very high; the most frequent peak results appear at 58.80 ± 12.94 nm, 53.00 ± 12.53 nm, 79.95 ± 12.72 nm, and 119.61 ± 11.87 nm for solutions at pH 8, 9, 10, and 12, respectively. However, radii sizes of 1259.83 ± 691.12 nm, 1447.85 ± 869.56 nm, 1195.50 ± 965.57 nm, and 577.50 ± 386.88 nm are obtained as the global average for NPs in PBS at pH 8, 9, 10, and 12 respectively. At physiological pH, both values are very similar, i.e., 144.10 ± 24.42 nm and 99.20 ± 50.21 nm are the radii results for the most frequent peak average and the global average of NPs size, respectively. When the pH drops to 3, the same behavior as pH 7.4 is observed, a behavior that could be attributed to repulsion forces happening between the NPs when the media is deprotonated, avoiding their aggregation. Having a look at the graph in Figure 3A, the distribution of NPs size at pH 3 is higher than at pH 7.4, meaning their size is more variable. However, if the value ranges from 5 to 7 again, aggregates are formed; in this case, the average radii size value is lower than the ones for NPs in alkaline pH values. The measured radii are 141.80 ± 30.17 nm, 197.90 ± 12.29 nm, and 139.40 ± 19.96 nm for solutions at pH 3, 5, and 7, respectively. The global average radii size is 138.40 ± 59.97 nm for pH 3, 499.23 ± 288.42 nm for pH 5, and 460.35 ± 236.60 nm for pH 7.

### 3.3. GDNF Release

ELISA results revealed a two-step GDNF release profile from the CMCht/PAMAM NPs, plotted in Figure 4. After 15 min, an initial fast release happens, being of 298.36 ± 3.14 pg mL^−1^. At 1 and 4 h and 2 days, the GDNF detected amount is 404.40 ± 19.95 pg mL^−1^, 411.73 ± 12.06 pg mL^−1^, and 418.50 ± 27.91 pg mL^−1^, respectively. At first, within the first 15 min, 30.07% of the GDNF is released, calculated against the release on day 30. After 1 and 4 h and 2 days, values continue increasing, but more slowly, the release is 40.76, 41.50, and 42.18%, respectively. Longer time points indicate a progressive and more slow-release up to day 30. On day 7, the amount released is 527.35 ± 21.23 pg mL^−1^. On day 14 is 755.78 ± 19.57 pg mL^−1^, and at the last time-point, on day 30, the detected concentration of released GDNF is 992.10 ± 22.22 pg mL^−1^.

### 3.4. In Vitro Cytotoxicity Assessment

The viability of iSCs incubated in the presence of the CMCht/PAMAM NPs and GDNF-loaded CMCht/PAMAM NPs is shown in Figure 5. On day 1, significant differences are seen between the control group and cells exposed to NPs. Dye reduction measurements by absorbance (au) are 6.00 ± 0.40, 3.78 ± 0.44, and 4.33 ± 0.44 for the groups of cells incubated with 0.1, 0.5, and 1 mg mL^−1^ of NPs, respectively. The measured absorbance increases to 12.65 ± 0.88, 41.33 ± 0.99, and 42.11 ± 0.48 when cells are incubated with GDNF-loaded NPs at concentrations of 0.1, 0.5, and 1 mg mL^−1^ of NPs, respectively. Control cells have resulted in a measured absorbance (au) of 10.22 ± 0.77, which represents higher cell viability if compared to the group of cells incubated with CMCht/PAMAM NPs, but lower if compared to the measured viability of cells incubated with GDNF-loaded CMCht/PAMAM NPs. On day 3, there are no significant differences between cell viability if the concentration of GDNF-loaded CMCht/PAMAM NPs is increased; the measured absorbance (au) is 113.56 ± 1.13, 114.22 ± 2.56, and 112.11 ± 2.59 for the concentration of 0.1, 0.5 and 1 mg mL^−1^ of NPs, respectively. However, if results are compared to cells incubated with 0.5 and 1 mg mL^−1^ of CMCht/PAMAM NPs, cell viability is significantly higher when the NPs are loaded with GDNF. For the control groups, the measured absorbance (au) is 85.67 ± 1.85, and for the groups of cells incubated with 0.1, 0.5, and 1 mg mL^−1^ of CMCht/PAMAM NPs is 108.00 ± 1.42, 101.22 ± 1.56 and 92.33 ± 0.11, respectively.

### 3.5. NPs Cellular Uptake

Imaging of labeled SH-SY5Y cells incubated for 24 and 72 h with FITC-CMCht/PAMAM dendrimer NPs and FITC-GDNF-loaded CMCht/PAMAM dendrimer NPs shows the effectiveness of their internalization into the cells. Figure 6A shows the internalization of the CMCht/PAMAM NPs and GDNF-loaded CMCht/PAMAM NPs. At 24 h, NPs appear to adhere to the cell membrane as they are dispersed around cells. Small clusters can be seen in the cell’s cytoplasm, which indicates the NPs are internalized by the cells after 72 h. The quantification of FITC-labelled CMCht/PAMAM NPs internalization is done by flow cytometry studies, revealing that after 24 h of SH-SY5Y cells incubation in the presence of FITC-conjugated CMCht/PAMAM NPs, the uptake is of 50.25 ± 1.96%. When NPs are loaded with GDNF, 55.87 ± 2.12% of the cells show fluorescence, reflecting an increase in the uptake. After 72 h, the difference is lower between the groups, i.e., 99.34 ± 0.45% and 99.97 ± 0.00% of the cells have uptake the CMCht/PAMAM NPs and the GDNF-loaded CMCht/PAMAM NPs, respectively. In Figure 6B, two representative graphs of the measurement of one sample at 24 h and another at 72 h are shown. The control refers to cells that have been incubated in a complete medium.

## 4. Discussion

The use of several types of NPs, such as silica-based, polymeric or magnetic, for neurotrophic factor encapsulation for PNR therapies is being widely investigated [24]. NGCs act as supports for PNR when nerve ends retract after an injury; they are tubular structures that can be tuned in order to enhance nerve regeneration through different techniques, such as neurotrophic factor incorporation, gifting the inner surface with topographical cues, or neural cell encapsulation [25]. The present work shows the synthesis of GDNF-encapsulated CMCht/PAMAM NPs as a nanotool for efficient GDNF delivery to treat peripheral nerve injuries. NPs size is well defined, and they are demonstrated to be stable in an aqueous solution due to the measured ζ-potential [26]. When NPs are loaded with GDNF, FE-SEM images reveal that NPs appear to be more compact and their shape is more well-defined, which could be due to the strong electrostatic interaction the GDNF has with the dendrimer NPs.

The Tg is low for CMCht/PAMAM NPs and GDNF-loaded CMCht/PAMAM NPs. These results indicate that both NPs possess a crystalline structure when used in the body. The Tg is lower when the NPs are loaded with GDNF, which is in concordance with DLS and FE-SEM results, which showed less polydisperse NPs size and more compact NPs when the GDNF is loaded, being these NPs more stable compared to the CMCht/PAMAM NPs. TCT results indicated that both NPs are thermally stable in the human body, and the high temperature that has to be reached to decompose the NPs could be due to the hydrogen bonds that are present in the PAMAM, as was previously pointed out elsewhere [27]. However, results differ from other studies of PAMAM NPs, where they decompose at a higher temperature [28]. This alteration could be a consequence of the CMCht grafting to the PAMAM, as it is demonstrated that the longer the grafted polymer, the lower the polymer’s stability [29]. As seen in FE-SEM images, the GDNF-loaded CMCht/PAMAM NPs appear to be more compact, which could be traduced into a more stable grafted polymer, as seen in DLS size results.

DLS measurements of the CMCht/PAMAM NPs and GDNF-loaded CMCht/PAMAM NPS in PBS changing the pH are performed to study their behavior in different environments that are present within the cells when they uptake external materials. At alkaline pH values and physiological pH, the average size is very high, being both values very similar. When the pH drops to 3, the same behavior as in pH 7.4 is observed. However, if the value ranges from 5 to 7 again, aggregates are formed, but in this case, the average radii size value is lower than the ones for NPs in alkaline pH values. The endosomal environment is acidic [30], meaning the NPs will aggregate while they are internalized and will later be released to the cell cytoplasm, desegregating [31]. 

The release profile of the GDNF is studied by ELISA and follows a two-step profile; an initial burst release is observed to be as late as 15 min after NPs are dissolved in PBS. After this initial fast release of the GF, a slower and time-lasting release is observed, which lasts at least 30 days. After a proximal nerve injury happens, the main limitation is the functional recovery, and the axonal regeneration is slow. In general, it takes around 1 month to regenerate 1 inch of a human peripheral nerve [32], so the CMCht/PAMAM NPs can perfectly serve as a booster to improve the regeneration rate, as it is seen that they at least release GDNF for one month. Cells could be exposed to a higher concentration of NPs, which would lead to a higher amount of released GDNF. In our case, the concentration used for cell culture is 5 mg mL^−1^, but this could be tuned once the NPs are included in NGCs for nerve grafting applications in order to achieve a therapeutic effect. It is already proposed a therapeutic window for exogenous GDNF administration for PNR purposes; a balance has to exist between the concentration and the duration of the treatment [33]. A low concentration of GDNF does not help axonal growth, but when levels are high, or the exposure to GDNF is for more extended periods, it does [34,35].

The in vitro cell culture studies have demonstrated the cytocompatibility of the NPs. On day 1, in the case of CMCht/PAMAM NPs without GDNF, the viability decreases, and on the contrary, when they have GDNF, the viability increases significantly. When iSCs are incubated for 3 days, the reduced viability of cells incubated with CMCht/PAMAM NPs is reversed, and cell viability gets improved. At both time points, if we compare cell viability regarding NPs concentration without GDNF, we see an undesired effect when the concentration is increased, 0.5 and 1 mg mL^−1^. However, no cytotoxic effect is observed; if the viability is compared to the control group, it is improved in the presence of the NPs after 3 days of culture. The viability is even more enhanced when NPs concentration is 0.5 and 1 mg mL^−1^, indicating a positive effect of the released GDNF on iSCs proliferation [36]. On day 3, there was no difference in the viability between the 3 concentrations of NPs with GDNF. There are no significant differences compared to iSCs incubated in a medium with 0.1 mg mL^−1^ of NPs without GDNF. These results suggest that iSCs growth is enhanced at the beginning, and in consequence, this could help achieve faster repair of the nerve [37]. Internalization studies have demonstrated that after 72 h of SH-SY5Y neural cells incubation with CMCht/PAMAM NPs and GDNF-loaded CMCht/PAMAM NPs, the uptake reaches almost the 100%. The fluorescence images confirm these results; there is no background fluorescence coming from the FITC-conjugated NPs, and all the visualized fluorescence is confined within the cells’ cytoplasm. Another fact that should be noted is that SH-SY5Y cell density is higher when incubated with GDNF-loaded CMCht/PAMAM NPs, which might be a consequence of the released GDNF [38].

## 5. Conclusions

The present work has demonstrated that it is possible to synthesize stable CMCht/PAMAM dendrimer NPs where the GDNF can be easily loaded following a simple protocol. Moreover, the GDNF is released fast at first, within 4 h, and then more slowly, which has been seen to impact cell viability after 24 h of incubation with the NPs containing the growth factor. The internalization study has revealed that after 72 h, almost 100% of the NPs are internalized by the SH-SY5Y neuronal cells. This high internalization rate makes the NPs beneficial for PNR purposes, as an initial boost of growth factor is required to promote regeneration. The following slow release of GDNF would help the complete repair of the nerve. In brief, the results are promising for further work related to the incorporation of the described NPs into nerve guidance conduits, which work as support to regenerate the injured nerves, guiding the regeneration from the distal to the proximal ends of the nerve. Further experiments should include in vivo assays to ensure biocompatibility and the effectiveness of peripheral nerve regeneration therapies.

## Figures and Tables

**Figure 1 pharmaceutics-14-02408-f001:**
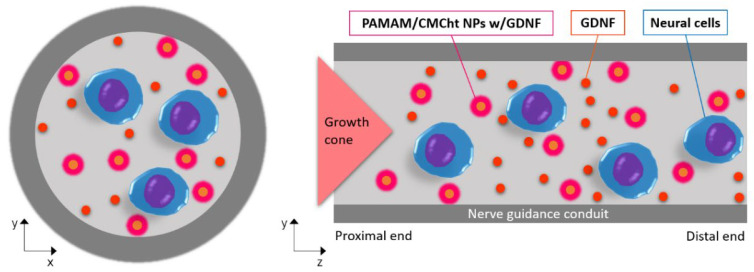
Illustration of NGC with the incorporation of CMCht/PAMAM NPs loaded with GDNF for GF release and improvement of PNR.

**Figure 2 pharmaceutics-14-02408-f002:**
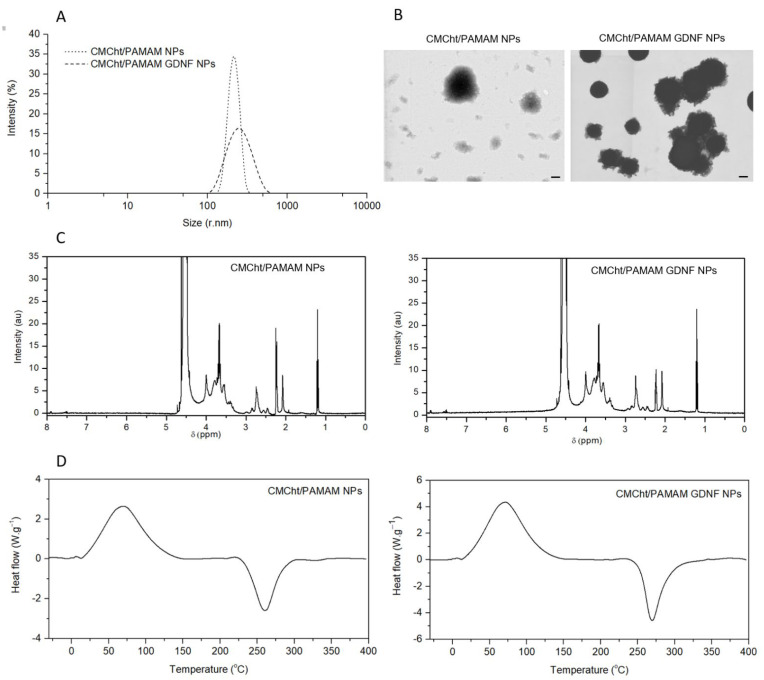
Characterization of CMCht/PAMAM NPs with GDNF. (**A**) Particle size distribution. (**B**) FE-SEM images of CMCht/PAMAM NPs without (**left**) and with (**right**) GDNF. Scale-bar = 100 nm. (**C**) H^1^NMR spectra of CMCht/PAMAM NPs without (**left**) and with (**right**) GDNF. (**D**) Differential scanning calorimetry spectrum of CMCht/PAMAM NPs without (**left**) and with (**right**) GDNF.

**Figure 3 pharmaceutics-14-02408-f003:**
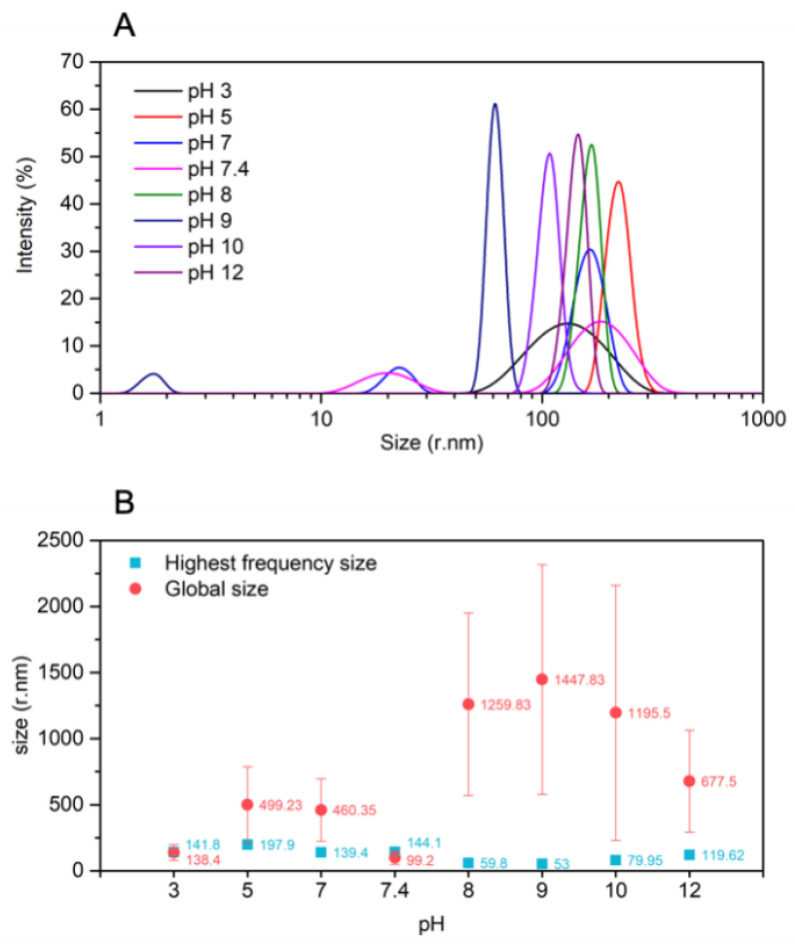
pH stability study of CMCht/PAMAM with GDNF NPs. (**A**) Size distribution from 1 to 1000 nm (radii) of CMCht/PAMAM NPs with GDNF at different pH values. (**B**) Highest frequency radii size value vs. global radii size average value.

**Figure 4 pharmaceutics-14-02408-f004:**
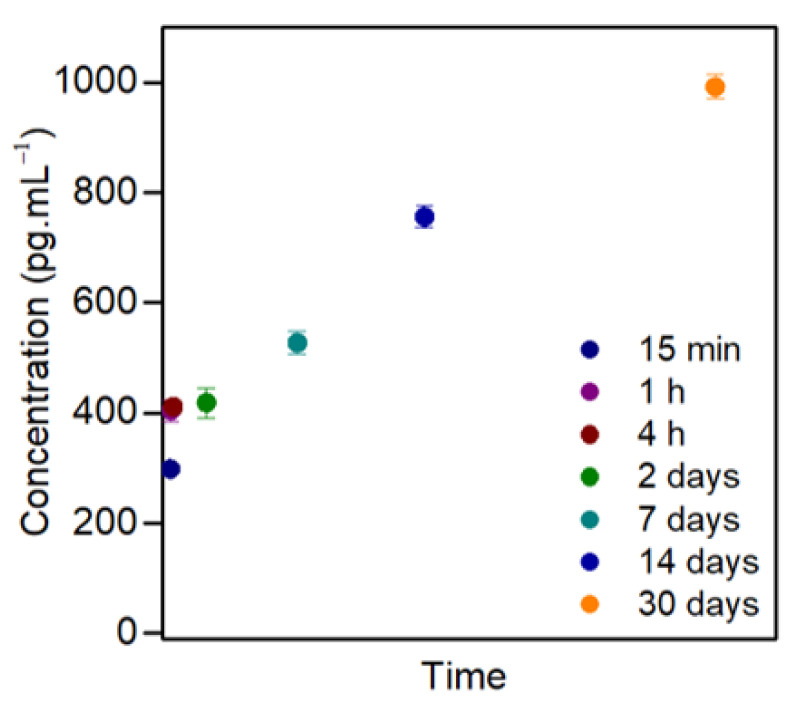
GDNF release profile from CMCht/PAMAM NPs quantified by ELISA at 15 min, 1 and 4 h, and 2, 7, 14 and 30 days. Quantitative data are presented as mean and error bars (*n* = 3).

**Figure 5 pharmaceutics-14-02408-f005:**
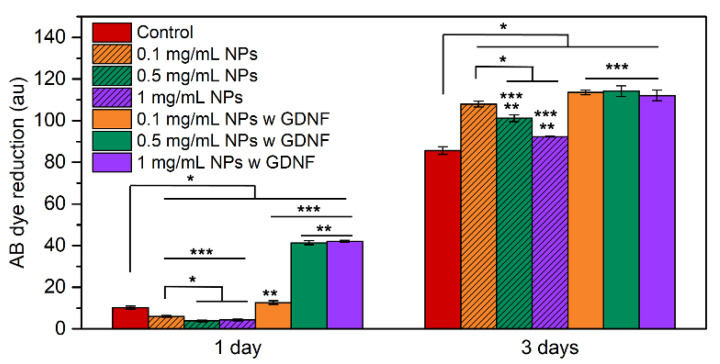
Viability AB results of iSCs in the presence of SCs cultured with CMCht/PAMAM NPs and GDNF-loaded CMCht/PAMAM. Quantitative data are presented as mean, and error bars (*n* = 3) and statistically significant differences are represented by *, ** and ***, where *p* < 0.05.

**Figure 6 pharmaceutics-14-02408-f006:**
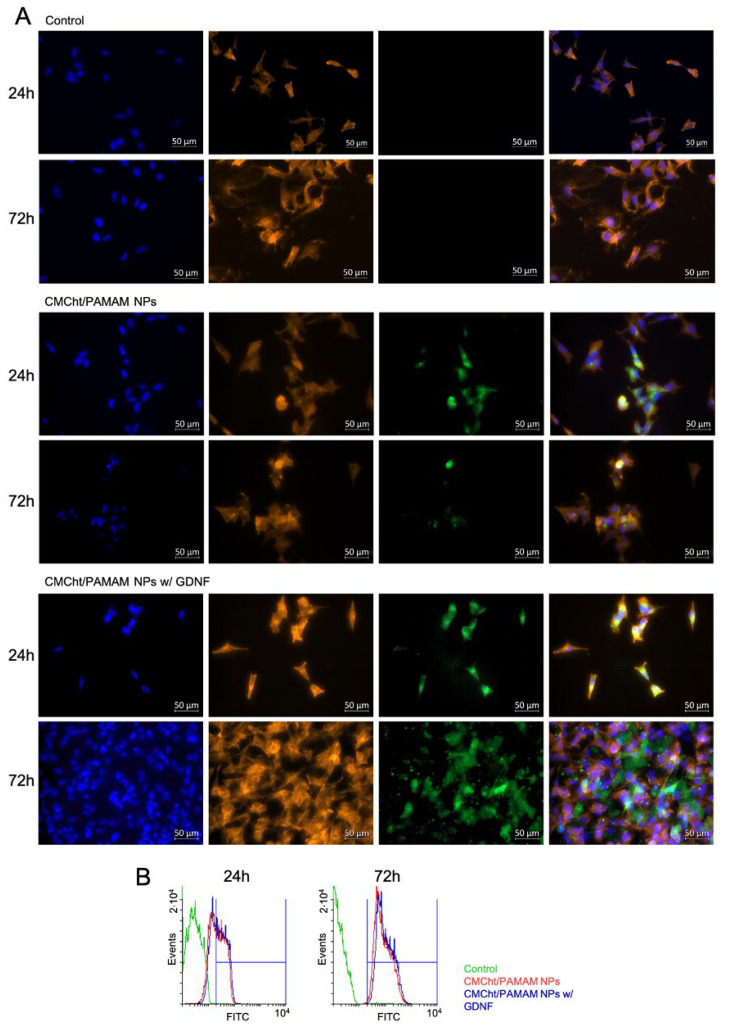
Uptake to SH-SY5Y cells of FITC-conjugated GDNF-loaded CMCht/PAMAM NPs. (**A**) Fluorescence images were taken at 24 h and 72 h after cells incubation in a complete medium, with the NPs and with the GDNF-loaded NPs, from top to bottom, and (**B**) Flow cytometry results at 24 h and 72 h.

## Data Availability

The data presented in this study are available on request from the corresponding author. The data are not publicly available due to privacy restrictions.

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
