# Peer review of "Glial Cell Line-Derived Neurotrophic Factor-Loaded CMCht/PAMAM Dendrimer Nanoparticles for Peripheral Nerve Repair"

_pharmaceutics, 2022, doi:10.3390/pharmaceutics14112408_

Round 1

Reviewer 1 Report

The methodology of characterization of nanoparticles lacks their concentration in water. This is a key parameter for zeta potential. The NMR method lacks the frequency with which the measurements were made (4000 or 5000HZ device). There is no indication of the degree of encapsulation of the sun with nanoparticles. After these additions have been made, the article may be published.

Author Response

We want to thanks the reviewers for the comments, and we are going to adress them in order to improve the manuscript.

The concentration of NPs used for the z-potential and dynamic light scatterng measurements is included in the materials and methos section, in page 2, and appears in red in the manuscript.

Regarding the frequency of the measurement in NMR, is described in page 2, in materials and method section as following: "the NMR spectra are obtained in a Mercury-400BB operating at a frequency of 399.9 MHz at 50 °C", highlighted in red. So, the frequency of the lamp in our measurements were of 399.9 MHz.

The way we determine the encapsulation of the growth factor (even if I am not sure that is the question about), was by mesuring the release at large time-points, thich is described in the results section in page 6. An ELISA test was made to determine this value.

Reviewer 2 Report

The authors synthesized carboxymethylchitosan/poly(amidoamine) (CMCht/PAMAM) dendrimer NPs loaded with GDNF for the exogenous administration of the GFs and also studies their physical properties and potential medical application.

1. The Figures from 2-5 are mistakenly labeled. Please check and correct.

2. In 3.2 NP stability, the authors did not explain the rationale of utilizing high frequency radii size to characterize radius of NP at different pH. It is also lack of scientific meaning of high frequency radius. Also from DLS, based on what physical parameters measured by DLS, the authors got the global radii and frequency radii ?

3. It is unclear that the rationale of using ELISA to characterize NPs. the authors need to add explanations to readers that why they believe ELISA is a good approach to characterize the NPs. Also there is no control plotted in the ELISA figure. it can not prove the specialty of syntheiszed NP on releasing.

4. Figure 4B in page 8, shows identical flow cytometry between NP and NP-GDNF. The authors need to provide controls on FTIC tagged NP with loaded GDNF; or using different fluorescent tags on NP and NP-GDNF to eliminate the suspicion that fluorescent labeling impact GDNF loading integrity. 

Author Response

We want to thank the reviewers for its comments and for giving us the chance to improve the manuscript. Regarding the specific comments, here go the answers:

1. The Figures from 2-5 are mistakenly labeled. Please check and correct.

We have checked and figures are properly labeled and mentioned in the menuscript.

2. In 3.2 NP stability, the authors did not explain the rationale of utilizing high frequency radii size to characterize radius of NP at different pH. It is also lack of scientific meaning of high frequency radius. Also from DLS, based on what physical parameters measured by DLS, the authors got the global radii and frequency radii ?

The global size is the average of all the size measurements did in a sample. This does not mean that a lot of the NPs have these sizes. We are usign the radii value that more is repeated in each samples. This means, there a are a lot of NPs with that value (the biggest part of them), but there are also some with bigger size, which might be aggregated NPs. Because of that, at physiological pH (7.4), both, the highest frequency size and the global size are very symmilar, meaning very few aggregation of NPs occurs. We are only showing these results to confirm the proper behaviour of the NPs at physiological pH.

3. It is unclear that the rationale of using ELISA to characterize NPs. the authors need to add explanations to readers that why they believe ELISA is a good approach to characterize the NPs. Also there is no control plotted in the ELISA figure. it can not prove the specialty of syntheiszed NP on releasing.

ELISA is a well known technique to detect proteins in solution, adding extrainformation on that will be explaining the basics of the technique. In our case, we used the specific kit for GDNF detection, we found that the most suitable way to do so. In the materials and methods section the way we proceeded is explained. It is highlighted on page 3.

 As control in this experiments, NPs without the GDNF was measured, but there is 0 detection of GDNF as expected. That was just done to confirm the specificity of the kit, which is very well characterized by the manufacturer. 

4. Figure 4B in page 8, shows identical flow cytometry between NP and NP-GDNF. The authors need to provide controls on FTIC tagged NP with loaded GDNF; or using different fluorescent tags on NP and NP-GDNF to eliminate the suspicion that fluorescent labeling impact GDNF loading integrity. 

The meaning that both uptakes are the same, means that the NPs are property internalized even with the presence of the GDNF. Which has only been shown to have a possitive effect on cell proliferation. I do not see the need to label with different tag as the experiments are performed separately, in different wells. We are just confirming the NPs uptake by cells at 24 and 72h. 

Reviewer 3 Report

In this manuscript, GDNF-loaded carboxymethylchitosan/Poly(amidoamine)Dendrimer Nanoparticles (CMCht/PAMAM NPs) used as a GDNF-release system to act on peripheral nerve regeneration has been investigated. This investigation is useful to develop method to repair peripheral nerve. The following suggestion should be considered.

1. In the introduction, authors pointed out that the ‘naked’ GFs administration has some drawbacks. However, some relevant literature studies on overcoming these shortcomings have not been reviewed. Especially, authors have published paper on “Macromol. Biosci.2010,10, 1130–1140”. “Carboxymethylchitosan/Poly(amidoamine)Dendrimer Nanoparticles in Central Nervous Systems-Regenerative Medicine: Effects on Neuron/Glial Cell Viability and Internalization Efficiency “. In the introduction, authors should evaluate the published papers and explain the innovation of this study.

2. In the discussion section, some results were not adequately explained. For example, (1) The reasons why the ζ-potential potentials are negative, and the difference in ζ-potential potential between A and B; the charge characteristics of GDNF should be stated. (2) Due to lack of thermal weight analysis results of the corresponding samples, the thermal effect peaks assigned to the DSC are not rigorous enough. (3) The specific physical implications of the highest frequent radii size and the average radii should be explained. Additionally, the large variability between the highest frequent radii size and the average radii should also be explained. In the text, the word, average radii, was used, however, in the note in Figure 1, the word, global radii size, was used. Why?

3. To better understand the rationale of the present study, a schematic of GDNF loaded by CMCht/PAMAM NPs should be offered.

Author Response

We want to that the reviewers for the comments, that have been useful to improve the manuscript.

Specific answers to the comments are the here:

1. In the introduction, authors pointed out that the ‘naked’ GFs administration has some drawbacks. However, some relevant literature studies on overcoming these shortcomings have not been reviewed. Especially, authors have published paper on “Macromol. Biosci.2010,10, 1130–1140”. “Carboxymethylchitosan/Poly(amidoamine)Dendrimer Nanoparticles in Central Nervous Systems-Regenerative Medicine: Effects on Neuron/Glial Cell Viability and Internalization Efficiency “. In the introduction, authors should evaluate the published papers and explain the innovation of this study.

The commented work has been included in the reference list. Many other works have also been published in the same topic (growth factors encapsulation to avoid their degradation), we had already mentioned these:

  1. Si, H.B.; Zeng, Y.; Lu, Y.R.; Cheng, J.Q.; Shen, B. Control-Released Basic Fibroblast Growth Factor-Loaded Poly-Lactic-Co-Glycolic Acid Microspheres Promote Sciatic Nerve Regeneration in Rats. Experimental and Therapeutic Medicine 2017, 13, 429–436, doi:10.3892/etm.2016.4013.
  2. Santos, D.; Giudetti, G.; Micera, S.; Navarro, X.; Del Valle, J. Focal Release of Neurotrophic Factors by Biodegradable Microspheres Enhance Motor and Sensory Axonal Regeneration in Vitro and in Vivo. Brain Research 2016, 1636, 93–106, doi:10.1016/j.brainres.2016.01.051.
  3. Mili, B.; Das, K.; Kumar, A.; Saxena, A.C.; Singh, P.; Ghosh, S.; Bag, S. Preparation of NGF Encapsulated Chitosan Nanoparticles and Its Evaluation on Neuronal Differentiation Potentiality of Canine Mesenchymal Stem Cells. Journal of Materials Science: Materials in Medicine 2018, 29, 4, doi:10.1007/s10856-017-6008-2.
  4. Marcus, M.; Skaat, H.; Alon, N.; Margel, S.; Shefi, O. NGF-Conjugated Iron Oxide Nanoparticles Promote Differentiation and Outgrowth of PC12 Cells. Nanoscale 2015, 7, 1058–1066, doi:10.1039/c4nr05193a.
  5. Giannaccini, M.; Calatayud, M.P.; Poggetti, A.; Corbianco, S.; Novelli, M.; Paoli, M.; Battistini, P.; Castagna, M.; Dente, L.; Parchi, P.; et al. Magnetic Nanoparticles for Efficient Delivery of Growth Factors: Stimulation of Peripheral Nerve Regeneration. Advanced Healthcare Materials 2017, 6, 1601429, doi:10.1002/adhm.201601429.

This is highlighted in page 1-2 in the introduction.

2. In the discussion section, some results were not adequately explained. For example, (1) The reasons why the ζ-potential potentials are negative, and the difference in ζ-potential potential between A and B; the charge characteristics of GDNF should be stated. (2) Due to lack of thermal weight analysis results of the corresponding samples, the thermal effect peaks assigned to the DSC are not rigorous enough. (3) The specific physical implications of the highest frequent radii size and the average radii should be explained. Additionally, the large variability between the highest frequent radii size and the average radii should also be explained. In the text, the word, average radii, was used, however, in the note in Figure 1, the word, global radii size, was used. Why?

(1) The explanation has been included in the manuscript in page 4 in the following way: "this decrease in the change might be due to the positive change of the GDNF, that reduces the overall negative ζ-potential of the NP".

(2) With the DSC analysis we were aiming to study the stability of the NPs, mostly because they are aimed to be used in vivo. The results are obtained are optimal for theis use. 

(3) The following explanation is in the text: "DLS measurements are performed (Figure 3A), and the highest frequent radii size and the average radii are plotted (Figure 3B) to evaluate their stability. In most cases, the highest frequent peak values, plotted in blue, are not similar to the global average, in red, which means large aggregates are formed with a large size that cannot be detected in the DLS. " This explains the meaning of both values, and also what is the explanation for the difference between them at some pH values. This description has been highlighted in red in the text (page 5). The terms "global average of NPs size" or "global average radii size" are used in the text, and  "global radii size average value" is used in the legend of Figure 2, all of them are meaning the same, the key word here is 'average' meaning we are taking all the measured values into account for the determination of the average radii size value. 

3. To better understand the rationale of the present study, a schematic of GDNF loaded by CMCht/PAMAM NPs should be offered.

We found this suggestion very interesting, and as there are already in previous works description about the CMCht/PAMAM NPs, we have in this case included an illustration in Figure 1 (page 2), to show how the NPs would be introduced in the nerve guidance conduit to promote pripheral nerve regeneration.

Round 2

Reviewer 1 Report

The article is ready for publication.

Author Response

We would like to thank the reviewers for its previous comments that made us improve the manuscript.

Reviewer 2 Report

The authors addressed my questions and comments. I recommend to accept with present form.

Author Response

We want to thank the reviewers comments.

Reviewer 3 Report

The manuscript has been modified according to the suggestions. However, there are also some problems in the revised manuscript. Some suggestions should be consideration.

1. For well understanding DSC results, thermogravimetric analysis (TG) of the corresponding sample should be carried out. Generally, the characteristic peaks in DSC curve may or may not be related to weight changes. TG analysis is very necessary to be carried out, especially for this investigation. In this study, the first small peak is attributed to the glass transition, the second peak is ascribed to the crystallization, and the third peak is put down to the thermal decomposition. According to the literature "Nanomedicine (Lond.) (2017) 12 (6), 581596", the conclusions mentioned in this study for first peak and the third peak are similar to that appeared in literature. However, in the literature, there was not the second peak appeared. In fact, it has been reported in the literature (ACS Appl. Mater. Interfaces 2019,11,4261642628), there was the loos water below 100 ℃ in TG curve. If there is water in the CMCht/PAMAM-GDNF, the peak at 100 ℃ appeared in DSC curve may be attributed to loss water rather than the crystallization. Therefore, TG of CMCht/PAMAM-GDNF should be carried out.

  2. As a scientific research paper, the results of the measured ζ potential and particle size involved in this investigation should be explained. In fact, the effects of pH on the charged features of CMCht/PAMAM and GDNF have been reported. For CMCht/PAMAM, the carboxyl can be protonated below pH=6, and the amino can be protonated below pH=2. (Advanced Functional Materials, 2008 18(12):1840-1853). In this investigation, pH = 2-6, there is not charge in CMCht/PAMAM, pH is above 6, negative charged CMCht/PAMAM appears, and negative charge can be enhanced with increase of pH. As a result, the size of CMCht/PAMAM becomes big with increase of pH due to the electrostatic repulsion. For GDNF, the isoelectric point is 9.26 Int J Nanomedicine. 2016; 11: 1383–1394). Namely, GDNF possesses positive charge as pH is below 9.26, and it has the negative charge as pH is above 9.26. Based on the information mentioned above, some results related to the measured ζ potentials and particle sizes should be explained, especially, for the results stated in line from 190 to 208. For example, the most frequent peak average (MFP) and the global average (GA) of NPs size at pH=3 is very similar to that at pH=7.4, at other pH, especially at pH from 8 to 12, there are obvious difference between MFP and GA. How can these results be interpreted. 

Author Response

We want to thank the reviewers for its comments, which are addressed here:

1- The peaks for the thermodynamic crystallization temperature (TCT) is of 91.18 and 70.78 °C for NPs without and with GDNF, respectively. These results are quite far from the 100°C of the water loss, or even if it is lower, as reported in the reference named by the reviewer. This is why, and as in the cited reference 'Nanomedicine (Lond.) (2017) 12 (6), 581596' states, this peak is attributed to the TCT.

2- As commented in the manuscript, the changes in pH can affect the aggregation of the NPs, that is why the most frequent peak average (MFP) and the global average (GA) of NPs are compared at each measured pH. In order to clarify the possible situation happening at pH3, the following sentence is included in the manuscript: 'When the pH drops to 3, the same behavior as pH 7.4 is observed, a behavior that could be attributed to repulsion forces happening between the NPs when the media is deprotonated, avoiding their aggregation. Having a look to the graph in Figure 3A, the distribution of NPs size at pH 3 is higher than at pH 7.4, meaning their size is more variable.'